# Lessons learned in the study of representational alignment in physical reasoning

**Felix J. Binder**
Department of Cognitive Science
University of California San Diego
fbinder@ucsd.edu

**Rahul Mysore Venkatesh**[†]**, Daniel L. K. Yamins**[*†]**, Judith E. Fan**[*]
[*]Department of Psychology, [†]Department of Computer Science
Stanford University
{rmvenkat,yamins,jefan}@stanford.edu

## ABSTRACT

Recent developments allow AI systems to perform cognitively complex and rich tasks. At the same time, collecting human behavior at scale is more feasible than ever. This convergence of trends allows for the combined large-scale study of human and AI behavior in rich domains and tasks. Such experiments promise to provide better insight into the representations and strategies underlying both human and AI behavior. However, doing so in a way that does justice to both humans and AI systems is challenging. Here, we outline the key considerations and challenges we've faced in a benchmarking study investigating physical understanding across humans and AI systems and discuss how we've addressed them.

## 1 INTRODUCTION

In recent years, the field of artificial intelligence (AI) has seen an explosion of increasingly performant models in domains of long-standing interest to cognitive science, including vision (Krizhevsky et al. (2017); Ramesh et al. (2021); Adamkiewicz et al. (2022), decision-making Mnih et al. (2015); Bakhtin et al. (2022); Chen et al. (2021)), and language (Ouyang et al. (2022); Chowdhery et al. (2022)). Such progress raises the possibility that these algorithms might embody dramatically improved cognitive models that succeed in describing human behavior in a wide range of complex, real-world settings. Reciprocally, systematic methods of evaluation pioneered in behavioral science will be of great value in identifying where AI models fail to match human performance levels or response patterns, clarifying the limits of existing technology and providing a guide for further AI advances.

This convergence of trends suggests a new wave of collaborations between cognitive science and AI: it allows for the design and deployment of systematic large-scale experiments to probe behavior in rich real-world domains and tasks that are of common interest in cognitive science and AI. Recent large-scale efforts have investigated the alignment of human and AI behavior in a range of domains, including vision (Hebart et al. (2019); Grauman et al. (2022)), language (Hu et al. (2023)), decision-making (Lin et al. (2023); Shu et al. (2021)) and social reasoning (Gandhi et al.).

Such efforts aim to provide a better insight into the representations and strategies underlying both human and AI behavior. However, collecting and comparing human and AI behavior is challenging. It requires the careful design of datasets, tasks and experimental designs in order to do justice to both human and AI cognition. Here, we outline key considerations and challenges we've faced in a benchmarking study investigating physical understanding across humans and AI systems (Bear et al. (2021)). While the particular decisions can vary across projects, we believe that the general structure of such an effort can be outlined.

## 2 KEY CONSIDERATIONS IN PHYSION

### 2.1 CHOOSING A DOMAIN

Which domains of activity are well-suited to compare the behavior of humans and AIs in a way that is mutually informative? One signal is an existing disagreement between different theories of

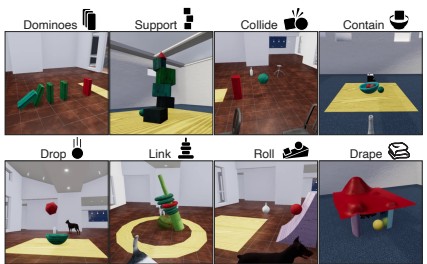

Figure 1: The Physion V1.0 dataset consists of short videos of physical interactions across 8 different interaction types. Each scenario type explores a different physical interaction of interest, such as causal chaining of events or objects supporting each other.

the mechanism underlying human behavior. Here, variation across the models can suggest a way to adjudicate between different theories. Another signal is a gap between human performance and the capabilities of current AI systems. Understanding how humans solve a task that is currently beyond the capabilities of AI systems can provide a roadmap for future AI research.

Physion investigates physical understanding of common sense physical interaction. Physical understanding is a core cognitive capability that is crucial for interacting with the world. Current AI systems still lag behind the flexibility and robustness of human physical understanding in daily life scenarios, even as computer vision has made great strides in recent years. At the same time, there is considerable disagreement about how humans solve physical understanding tasks. Some accounts argue that humans use an object-oriented representation to simulate physical interactions in their environment similarly to a video game engine (Battaglia et al. (2013); Ullman et al. (2017)), while others argue that a set of heuristics is used instead (Ludwin-Peery et al. (2021)).

## 2.2 DATASET

A crucial component is the dataset used for training and behavioral evaluation. For many domains, the cognitive science literature can provide a starting point for identifying interesting and relevant phenomena that should be included in the dataset. A dataset whose axes of variation track cognitively interesting phenomena promises to elicit more informative behavior. Different items from a dataset should cover a wide range of difficulty, both for humans and AI models. In particular, the dataset should include adversarial examples: items for which humans or AI models perform reliably below chance. By probing the particular errors that humans and AI systems make, it is possible to gain insight into the underlying representations and strategies.

For Physion, we developed the Physion V1.0 dataset. The dataset consists of videos of physical interactions in a three-dimensional environment. The physical interactions are grouped into 8 different scenario types, each of which explores a different physical interaction of interest drawn from previous cognitive science and developmental psychology literature, such as causal chaining of events or objects supporting each other. The videos were generated using TDW (Gan et al. (2020)). During the design of the dataset, informal piloting was performed to create a set of videos for which people made the correct prediction halfway between chance (50%) and ceiling performance (100%): in other words, the dataset was designed to have an average performance of 75% accuracy across the dataset, but with a wide range of the average correctness for individual video clips.

## 2.3 DESIGNING TASKS TO ELICIT HUMAN & AI BEHAVIOR

Humans and AI systems are embodied in different ways and have very different capabilities for behaviors. Understanding and accounting for the difference between the inputs and actions available to humans and AI systems is crucial for designing a task that is fair for both. A well-designed task allows a system with the right sort of representation to perform well. While deep learning systems excel at prediction and classification tasks, choosing richer actions in service of a goal is a harder domain for AI systems. Allowing AI systems to take actions requires the design of particular interfaces and the training of the model to make use of it. Especially if this requires reinforcement learning, this can be challenging. Humans on the other hand excel at interventions (such as placing an object to block a physical interactions), but struggle on other tasks. For instance, while eliciting

generative outputs from a suitably designed model (such as the synthesized next frame of a video) can be easily done for AI systems, such generation is nearly impossible for humans. Likewise, humans are poorly numerically calibrated and struggle to accurately report a calibrated confidence in their predictions, even on tasks that they are capable of solving (Liberman & Tversky (1993)).

To elicit physical understanding from human and AI, we designed the Object Contact Prediction (OCP) task. In this task, two objects in a scene are highlighted. The task is to predict whether they will come into contact with each other. The beginning of the interaction are shown (such as a ball rolling towards a tower, cutting out before the impact), and based on those initial frames, the participant or model has to predict whether in the rest of the video the two objects will touch. The dataset was created so that in 50% of the cases the two objects will come contact each other. This task is designed so that it is easily solved provided a good representations of the future state.

## 2.4 COLLECTING HUMAN DATA AT SCALE

An effort of this sort requires collecting human responses to hundreds or thousands of items, making the collection of human data at scale crucial. While in-person collection of data is possible, it is time-consuming and expensive. Rather, the collection of human data at scale is typically done online. For Physion, human data was collected using Prolific. For each of the 8 scenario types, we collected data from 100 participants. To support the data collection effort, we developed a web-based experiment system for collecting human behavior on cognitive AI benchmarking tasks. The framework consists of a JsPsych-based frontend (de Leeuw et al. (2023)), a backend with a database holding experiment data and a connection to a content delivery network (such as Amazon S3) for serving the dataset itself. In addition, code for creating and analyzing experiments is provided. The framework is designed to be easily deployable and customizable for other studies that compare human and AI behavior and can be accessed here: `https://github.com/cogtoolslab/cognitive-ai-benchmarking`.

## 2.5 BUILDING INFORMATIVE AI MODELS

While AI systems are not explicitly designed to be models of human cognition, they can still be informative regarding human cognition. When comparing AI models to human behavior, which models are most fruitful to include? The axes of variation in the models can correspond to different hypotheses about how humans are solving the task. The models that were benchmarked in Physion followed a structured setup, with different models embodying different variations across this structure—see figure 2.

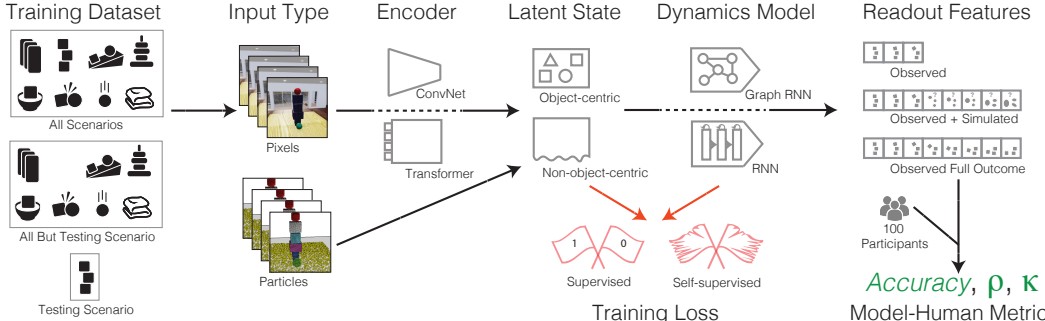

Figure 2: The structured setup of the model consists of an encoding layer that takes in visual information or a physical ground truth representation, an encoding module that represents the scene, a dynamics model that predicts how the scene evolves over time and a readout layer that produces a response. This structure allows for the structured variation of a range of features, such as input format, encoder types, supervision signal and for lesions such as leaving out the dynamics module.

**Training Data** What is the distribution of data the models are trained on? By selecting the training data of the model, we can vary the information that the model receives. In Physion, the dataset was grouped into 8 scenario types, each of which explores a different physical interaction. We trained the

models either on all scenarios, all but the tested on or only on a single scenario type. This allowed us to investigate the generalization of the models across different physical interactions.

**Input Format**   In what format is the input fed into the model? Changing the modality of the input can stand in for different hypotheses about the nature of human representations. In Physion, most models received pixel-based input as a sequence of images. If such a model performs poorly, it could either be due to the difficulty of extracting the relevant information from the pixel-based input, or due to the difficulty of predicting the future state of the scene. To isolate these two factors, we also included models that took as input a particle-based representation of the scene, which gives ground-truth information about the shape, location and size of the objects in the scene.

**Encoder**   The input is passed into an Encoder, which computes a latent state representing the input. This corresponds to the representation that the model is using in further steps to arrive at its output. The architecture of the encoding layer can vary—for example, in Physion we considered models with both transformer and convolutional architectures. Since the output of the encoder is not directly connected to the task, encoders that have been trained on different datasets can be used.

**Supervision Signal**   The output of the encoder is a latent state which represents the input. How should this latent state be trained? The supervision signal used to train the encoder determines the format of the representation that the model learns. A self-supervised training signal such as a pixel-level reconstruction loss in a video prediction task might lead to fundamentally different representations than using ground truth information about the physical state of the scene as training signal. This allows for the variation of the representational content of the latent state, which leads to a certain pattern of behavior in the context of the whole model. Here, different supervision signals on the latent state can be understood as standing in for different hypotheses about the nature of representations in human physical reasoning: are humans representing their environment in a compositional and object-oriented way, or does physical reasoning rely on a different kind of representation?

**Dynamics Model**   In order to predict how the interactions between objects evolve, the latent state is fed into a dynamics model. The dynamics model predicts how the scene evolves over time and outputs a representation of the future state of the scene. Physion incorporated different architectures for the dynamics model, including recurrent neural networks and graph RNNs. The dynamics model is supervised by a similar supervision signal as the encoder.

**Readout Layer**   Humans are task generalists whose representations are learned through a lifetime of varied experiences and used across a wide range of tasks. Training powerful AI systems requires large amounts of data and computing resources. Ideally, the representations of an AI are not only trained for one specific task, but are general. How can we extract behavior from the representation of a model? A shallow readout layer can be trained on top of the representations to perform the task. In other words, the latent state is fed into a linear layer that is trained on the particular task. This way, the model can be trained to do the task without having to train the entire model from scratch. This allows a given model to be quickly adapted to new tasks. In Physion, the readout layer was trained on the OCP task to pull the relevant information from the representation in the latent state (namely, whether the two objects are touching). The readout layer can also be applied to the output of the encoder, thereby lesioning the dynamics model. By training and testing the readout layer on the observed representation without dynamics, we isolate whether the relevant features for the task are already present in the encoded state without explicit forward simulation. Likewise, by feeding the entire video including the future frames into the encoder, we can ensure that the simulated future state would actually contain the information necessary for the task, assuming that the dynamics model correctly predicts the interactions. Decomposing task-general representations from the task-specific readout layer allows both for flexibility across tasks and the ability to investigate the contribution of different parts of the model on the behavior of the model.

## 2.6   RELATING HUMAN & AI BEHAVIOR

When comparing human and AI behaviors on a given task, there are two broad avenues of comparison: absolute performance, and the similarity and differences of AI behavior to that of humans. The absolute performance of humans on a given task is an important benchmark for the development of

AI systems. It provides a lower bound of the performance that is possible given the particular input on a task. It can be used to pinpoint where AI systems fall short. At the same time, humans are fallible and make errors. The particular error patterns in human behavior can be instructive. Unlike models, humans are trained on a wide range of tasks and have a lifetime of experience. This allows for a flexibility in behavior as well as a robustness to certain adversarial inputs that is unmatched by AI systems. Generally speaking, humans have high performance on most cognitive tasks, and the higher the absolute performance of AI systems, the more they act like humans. But it is worth disentangling the absolute performance and the particular patterns of errors that humans and AI systems make. Similar error patterns between humans and AI systems can indicate that both are using similar representations and strategies to solve a task. Knowing about the divergence of human and AI behavior allows for the training of AI systems to match human behavior more closely. While this might reduce the absolute performance of the AI system on a particular task, it can serve to make the AI system more robust and flexible by aligning how information in the system is represented and processed.

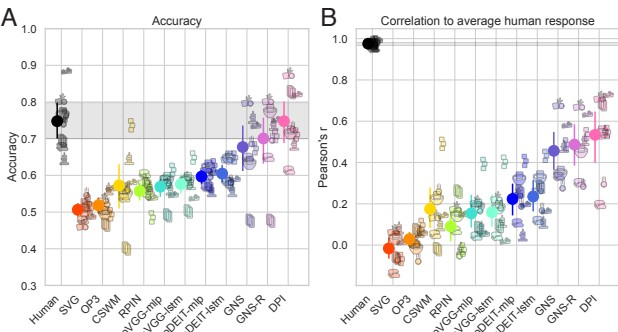

Figure 3: (A) Absolute performance (rate of correct predictions) is shown. The gray band indicates the performance of humans on the 8 scenario types (shown with icons). Most models fall short of human performance, but model that receive physical ground truth approximate human performance. (B) The correlation of model predictions to human behavior. Even models that achieve human-level performance behave differently. "Human" shows the correlation of split halves of human responses with each other, indicating that human behavioral patterns are robust. Error bars are 95% confidence intervals.

In Physion, three sources of data were compared: the predictions of humans, the predictions of AI models, and the ground truth outcome for a given video. We find that humans perform well on the OCP task, with an average accuracy of about 75% accuracy—see figure 3. Most models perform noticeably worse. Self-supervised models that receive pixel-level supervision perform the worst, while models that receive object-level supervision perform better. This suggests that object-centric representations are helpful for the task. However, even models with object-centric representations fall short of human performance. Models that receive ground truth physical data as inputs achieve near-human performance, suggesting that the bottleneck is getting a good representation of a scene from vision rather than learning physical dynamics. However, when comparing the pattern of responses of the models to human behavior, we find that even models that achieve human-level performance behave differently. This suggests that the models are using different representations and strategies to solve the task.

## 3 DISCUSSION

We have outlined the key decisions we faced in a study investigating physical understanding in humans and AI: choosing a domain, designing a dataset, developing a task, collecting human data at scale, building informative AI models and relating human and AI behavior to gain insight into the underlying representations and strategies. This template is generative: for each of these steps, different choices can be swapped in. For example, the OCP task could be exchanged for a task that elicits richer behavior than just a binary prediction (such as one that predicts the particular location of the contact). In a concrete example, (Martinez et al. (2023)) used the framework of Physion to ask for how interesting a given video is, comparing different accounts of curiosity in AI to what people find interesting.

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
