# OpenReview forum: "Lessons learned in the study of representational alignment in physical reasoning"
_ICLR.cc/2024/Workshop/Re-Align — ICLR 2024 Workshop Re-Align Poster_

### Official Review · Reviewer_Q4z8 · 2024-02-23
**Interesting paper that perhaps sells itself a bit too short**

**Rating:** 2
**Fit:** 2
**Confidence:** 2

**Workshop Review:**

Strengths:
+ Interesting motivation: study investigating physical understanding across humans and AI systems
+ The Physion V1.0 dataset and Object Contact Prediction (OCP) task are interesting contributions! The results from the Prolific data collection are also interesting. I wish they were all mentioned as contributions in the introduction. I think the abstract and intro are selling the paper contributions short; there’s a lot more going on here that’s meaningful and interesting than just outlining decisions for how to design the study!

Weaknesses:
- The contributions are stated relatively vaguely in the abstract and intro (“Here, we outline key considerations and challenges we’ve faced in a benchmarking study investigating physical understanding across humans and AI systems”). I believe the manuscript could benefit from more specificity here. As I said above, I think the paper sells itself short in what it’s achieved. I would revise the abstract and intro to more clearly tell the story of how the study was designed and what observations we can glean from it.
- “representation” is typically an overloaded term in the community. This paper didn’t make it clear what a representation is to them, and this lack of specificity makes it unclear to me whether this is a good fit for the workshop.
- I would’ve liked a stronger description of prior work; in the current form, it’s hard to tell what prior work has done in this space and how this work is different from that.

**Reason For Not Giving Higher Score:**

The fit is a bit unclear to me. I don't really understand what the authors define as representation here.

**Reason For Not Giving Lower Score:**

There's still an interesting study here that could be of interest to the community.

**Reviewer Domain:**

machine learning

---

### Official Review · Reviewer_jXjF · 2024-02-24

**Rating:** 1
**Fit:** 2
**Confidence:** 2

**Workshop Review:**

This paper summarizes the authors' challenges and things they considered when trying to create a human-AI benchmarking task.

Some strengths include the framing of a human-AI challenge in explicit terms of benchmarking, and the discussion of the two types of AI performance metrics ("absolute performance" and behavior).
Some weaknesses include the confusing overall narrative of the paper, not getting across the salient points, and not clearly scoping the paper. It would be helpful to flag exactly what the paper contributes early on.

Examples of sentences that would benefit from more clarity (these are just representative examples):

"Generally speaking, humans have high performance on most cognitive tasks, and
the higher the absolute performance of AI systems, the more they act like humans." -- this is highly controversial, and doesn't include any background material/literature. The results given on the last page also seem to contradict this entirely?

"in other words, the dataset was designed to have an average performance of 75% accuracy across the
dataset, but with a wide range of the average correctness for individual video clips." -- unclear what you are trying to say, maybe an illustration or example would help?

**Reason For Not Giving Higher Score:**

It's a bit unclear/difficult to understand the salient points of the paper; if the idea is simply to discuss potential considerations, a much tighter coupling to the related work/existing literature would be very beneficial and would greatly strengthen the narrative.

A "contributions" section would greatly improve the paper. A clear goal for the paper would also help.

There are also several proofreading issues that start to become distracting, and I think the paper would benefit from some review/polishing rounds for readability. For example, all citations that are supposed to be in parenthesis are instead floating in the text.

I'd also recommend multiple rounds of reviews for readability and flow;  the sentence structure feels a bit odd in many places, but I'm sure this can be improved given time and a few cycles of feedback.

The ending is very, very abrupt. Perhaps you intended to make the discussion section more than a single paragraph and something got cut off?

**Reason For Not Giving Lower Score:**

N/A

**Reviewer Domain:**

machine learning

---

### Decision · Program_Chairs · 2024-03-02

Accept (Poster)